# Deep Learning Segmentation in 2D X-ray Images and Non-Rigid Registration in Multi-Modality Images of Coronary Arteries

**DOI:** 10.3390/diagnostics12040778

**Published:** 2022-03-22

**Authors:** Taeyong Park, Seungwoo Khang, Heeryeol Jeong, Kyoyeong Koo, Jeongjin Lee, Juneseuk Shin, Ho Chul Kang

**Affiliations:** 1Department of Biomedical Informatics, Hallym University Medical Center, 22 Gwanpyeong-ro, 170 beon-gil, Dongan-gu, Anyang-si 14068, Gyeonggi-do, Korea; taeyong@mach.hallym.or.kr; 2School of Computer Science and Engineering, Soongsil University, 369 Sangdo-ro, Dongjak-Gu, Seoul 06978, Gyeonggi-do, Korea; swkhang@naver.com (S.K.); heeya8876@naver.com (H.J.); koomaro@naver.com (K.K.); leejeongjin@ssu.ac.kr (J.L.); 3Department of Systems Management Engineering, Sungkyunkwan University, 2066 Seobu-ro, Jangan-gu, Suwon-si 16419, Gyeonggi-do, Korea; jsshin@skku.edu; 4Department of Media Technology & Media Contents, The Catholic University of Korea, 43 Jibong-ro, Bucheon-si 14662, Gyeonggi-do, Korea

**Keywords:** percutaneous coronary intervention, image segmentation, convolutional neural network, nonrigid registration, multimodality registration

## Abstract

X-ray angiography is commonly used in the diagnosis and treatment of coronary artery disease with the advantage of visualization of the inside of blood vessels in real-time. However, it has several disadvantages that occur in the acquisition process, which causes inconvenience and difficulty. Here, we propose a novel segmentation and nonrigid registration method to provide useful real-time assistive images and information. A convolutional neural network is used for the segmentation of coronary arteries in 2D X-ray angiography acquired from various angles in real-time. To compensate for errors that occur during the 2D X-ray angiography acquisition process, 3D CT angiography is used to analyze the topological structure. A novel energy function-based 3D deformation and optimization is utilized to implement real-time registration. We evaluated the proposed method for 50 series from 38 patients by comparing the ground truth. The proposed segmentation method showed that Precision, Recall, and F1 score were 0.7563, 0.6922, and 0.7176 for all vessels, 0.8542, 0.6003, and 0.7035 for markers, and 0.8897, 0.6389, and 0.7386 for bifurcation points, respectively. In the nonrigid registration method, the average distance of 0.8705, 1.06, and 1. 5706 mm for all vessels, markers, and bifurcation points was achieved. The overall process execution time was 0.179 s.

## 1. Introduction

Recently, cases exhibiting the coronary artery disease have significantly increased owing to the extension of the average life expectancy and lack of exercises [1,2]. The coronary artery disease is caused by the narrowing or closing of the coronary artery due to the stenosis and metabolic failure of a heart muscle [3]. Percutaneous coronary intervention (PCI) is one of the frequently used treatment methods for coronary artery disease and demonstrates advantages for a patient such as the minimization of mental and physical strain [4,5]. However, as this treatment depends on 2D X-ray angiogram (XA) images and anatomical 3D vascular structures, which are understood primarily by medical doctors through intuition and haptic feedback, the accuracy of this difficult medical procedure is not guaranteed [6]. While 2D XA images enable the identification of the movement of blood vessels in real-time, they demonstrate a limitation as the information on the structure of 3D shapes is lost in the process of being projected onto a 2D plane. 3D-computed tomography angiography (CTA) images are primarily images captured at the end-diastole or end-systole; consequently, there are limitations in representing the movement of blood vessels in real-time. Therefore, research on assistive technology that can represent moving blood vessels in real-time is increasing.

To segment coronary arteries in XA images, various methods [7,8,9,10] have been proposed. Wang et al. [7] proposed a method using hessian matrix multi-scale filtering and multi-seed region-growing. In this method, two algorithms were combined and applied in consideration of the characteristics of XA images to obtain robust results. Zhou et al. [8] proposed a multi-feature based fuzzy recognition method. For robust vascular tree structure extraction, a probability tracking model that analyzed prior knowledge of features of XA images was used. Lee et al. [9] proposed an intensity-vesselness gaussian mixture model method in which the intensity value and vesselness value of the XA images were combined. In addition, spatial prior probabilities for foreground and background were additionally used. However, since these methods are performed only by 2D XA images, it is difficult to grasp the information due to the 2D projection of 3D information, and there are disadvantages in that the structure cannot be accurately identified due to overlap or crossing exists between vessels and background clutter of organs, bones, and intervention tools. Sun et al. [10] proposed a model-guided approach that additionally utilizes 3D blood vessel information. However, since only the average distance is used in the segmentation process, there is a limit to the use of 3D blood vessel information, which has a disadvantage in that segmentation accuracy is lowered. These classical image processing methods have the drawback that they take a very long time to execute in combination with various filters or probabilistic models. In addition, it is necessary to define knowledge-based parameters that are limited to a specific data group, or semi-automatically through manual seed point input is required. As a result, these approaches cause limitations due to the difficulty of generalization of target features and real-time application.

Extensive research [11,12,13,14,15,16,17] has been performed on multimodality registration of coronary arteries. Metz et al. [11,12] and Baka et al. [13] proposed a registration method between 4D CTA and XA images. Metz et al. [11,12] was used for registration by generating digitally reconstructed radiography images from 4D CTA images taken in multiphase in order to minimize the difference in the deformation of vessels due to the heartbeat. Baka et al. [13] proposed a nonrigid registration using an oriented gaussian mixture model between the centerline extracted from the XA and CTA images. For the centerline, a 4D gaussian mixture model (GMM) was constructed by adding a direction vector, and a statistical shape model and modified L2 distance measurement were used. Kaila et al. [14] and Rivest-Hénault et al. [15] proposed a registration method between 3D CTA and Bi-plane XA images. Kaila et al. [14] proposed a coherence point drift registration that constructs the 3D centerlines from Bi-plane XA and CTA images as a GMM and aligns it to the point where the posterior probability is maximized. Rivest-Hénault et al. [15] performed global registration to the point where distance is minimized through affine transformation. In the local deformation, an energy term was designed and applied in consideration of the characteristics of blood vessels and myocardium. However, since these methods use 4D CTA or Bi-plane XA image information, they cannot be performed on general 3D CTA and XA images, and limited use is possible. Kim et al. [16,17] extracted centerlines of blood vessels from XA and CTA images, and projected 3D blood vessels as 2D blood vessels to perform registration based on thin plate spline robust point matching algorithm [18]. However, the topology information of the 3D CTA blood vessels is not preserved in this method because the transformation of the 3D objects is not considered. This is because the registration process is performed only in 2D after the centerlines of the 3D CTA blood vessels are projected as 2D images. In addition, owing to the lack of depth information, the convergence to the local minimum in areas where local errors exist may lead to wrong registration results.

To solve these problems, this paper proposes a fast, robust, and accurate convolutional neural network (CNN)-based segmentation and nonrigid registration method based on phase structure analysis and similarity comparison. Unlike classical image processing methods, a CNN-based coronary arteries segmentation method does not require specific parameters according to the input data, and a robust fully-automatic segmentation result can be obtained. The phase structure analysis performs a comparative analysis with the phase structure of 3D blood vessels for the purpose of compensating for the insufficient 2D XA image information. It uses a graph structure to reduce the complexity of blood vessels and achieves efficient comparative analysis; moreover, this method performs reconstruction and comparative analysis of the entire structure by assuming all possible connection paths beginning from the smallest segment unit. During this process, point matching between two vessels is performed based on the result of rigid registration to verify and apply accurate correspondences. In this study, for more accurate nonrigid registration, similarity comparison between corresponding blood vessels is performed using graph structures. The similarity comparison uses feature information such as the curvature, thickness, and shape of blood vessels. In addition, as energy functions are defined and utilized based on the blood vessel feature information, two vessels can be matched to the optimal position.

## 2. Methodology

In this paper, a fast, robust, and accurate multimodality segmentation and registration method using the process shown in Figure 1 was proposed to match the blood vessels in 3D CTA images, which were obtained before the procedure, and 2D XA images, which were obtained during the procedure from the same patient, to provide useful guiding information to the operator.

In this paper, we used the characteristic information of blood vessels for application in real time. To effectively utilize the blood vessel information of the preoperative 3D CTA images, the 3D centerline and phase structure extracted from the 3D CTA images [19,20] were used. The 2D XA images acquired in real time during the procedure were segmented and reconstructed using CNN and 3D information. Rigid registration was used to utilize the data of the same patient with information of different modalities [21]. Thereafter, the phase structures of the two blood vessels were compared and analyzed based on the results of the rigid registration. Graph structures were generated and used for efficient calculation and comparative analysis. Although the phase structure of 3D blood vessels was clearly divided based on the bifurcation point, the 2D blood vessels contain certain inaccurate information. To improve the 2D information, all possible connection paths were assumed to reconstruct the connection structure; then, the connection relationships of the two vessels were compared and analyzed. Although the graph structure was efficient for computation and comparative analysis, as the complexity of blood vessels were reduced to simplify the information regarding the blood vessels, it lacked feature information such as the curvature, thickness, and shape of blood vessels. Therefore, similarity comparison between the pieces of feature information of the two vessels was performed. The matching relationship between the two vessels for comparison of the phase structure and similarity was obtained by performing a slope-based point matching; moreover, an energy function was defined to change the position and shape of the 3D centerline and was optimized to a minimum value.

### 2.1. CNN-Based Coronary Artery Segmentation

In this paper, we used CNNs for segmentation of coronary arteries in 2D XA images. The ground truth for learning was created by a cardiologist and researcher with more than three years of experience using the semi-automated blood vessel segmentation method [22] and Adobe Photoshop CS3.

#### 2.1.1. Base Architecture

We used the network as shown in Figure 2 based on EfficientNet [23] and U-Net [24] for CNNs segmentation of coronary arteries from 2D XA images. The network consists of an encoding path and a decoding path. In the encoding path, it is a structure with maximum accuracy within the constraints of expression capacity and computational complexity through compound scaling, and the prediction accuracy was improved with much fewer parameters and FLOPS through optimization. In the expanding path, it fuses coarse semantic and fine appearance information using the information of the previous layer to make sophisticated predictions. The MBConv block consisted of inverted residual block and SE block. The inverted residual block represents a bottleneck block structure that increased the number of channels through the expansion layer and decreased it to the number of input channels through the depth-wise convolution and projection layer. The SE block uses global average pooling (GAP) to squeeze important information from each channel and performs recalibration according to the importance of the channel through channel-wise dependencies calculation.

#### 2.1.2. Loss Function

In order to reflect the characteristics of blood vessels, it learns through the loss metric considering each pixel and the shape. The overall loss function can be defined as:(1)$$L=L_{BCE}+L_{shape}$$
where $L_{BCE}$ represents pixel-wise binary cross-entropy loss used for binary segmentation to distinguish between background and blood vessels.
(2)$$L_{BCE}=-\frac{1}{K}\overset{K}{\underset{k=1}{\sum }}\left(y_{g_{k}}\log\left(x_{p_{k}}\right)+\left(1-y_{g_{k}}\right)\log\left(1-x_{p_{k}}\right)\right)$$
where K is the number of pixels of the maps, $y_{g}\in \left[0,1\right]$, $x_{p}\in \left[0,1\right]$ denotes the label map (training data of ground truth) and the predicted map. The pixel-wise binary cross-entropy loss lacks reflection of features such as the overall shape of the segmentation object. An additional $L_{shape}$ penalty term was used to take into account elongated shapes such as conduits of blood vessels.
(3)$$L_{shape}=\frac{1}{K}\underset{k}{\sum }\sqrt{\underset{i}{\sum }{\left(y_{d}x_{e}\right)}^{2}+\epsilon^{2}}$$
where $y_{d}\in \left[0,1\right]$ is the distance map generated by the chamfer distance [25] for the edge of $y_{g}$, $x_{e}$ denotes the edge created from $x_{p}$. $\epsilon =10^{-7}$ is a small number for numerical stability.

#### 2.1.3. Learning the Network

The training data used a semi-automatically generated 344 Case, which had a resolution of 512 × 512 pixels and pixel spacing values in a range of approximately 0.258 to 0.293. To improve segmentation performance, rigid transformation, crop, normalization, gaussian noise, gaussian smoothing, and cutout [26,27] methods were randomly combined and applied as an augmentation for learning the network. The cutout image augmentation was applied to reflect the loss of information such as crossing or overlapping during the projection process and narrowing or occlusion of blood vessel areas. Xavier uniform initialization [28] was used to initialize the weights of the network. The Adam Optimizer was used, and the learning rate was 3 × 10^−4^.

### 2.2. Multimodality Rigid Registration

The position and shape of blood vessels in preoperative 3D CTA and intraoperative 2D XA images of the same patient may differ due to other external factors such as the patient’s posture, heart rate, and respiration, and differences in protocols between imaging devices. In order to reduce these errors, registration was performed at the position where distance difference between the 3D and 2D centerline is the smallest by using the Euclidean distance through the optimization of the transformation function [21]. It is based on Digital Imaging and Communications in Medicine (DICOM) information obtained from the C-arm, which is the imaging apparatus of the 2D XA image, to reduce the search space for registration. The DICOM information is applied to 3D centerline extracted from 3D CTA image. Table 1 shows DICOM information.

The transformation function consists of a translation vector Tx, Ty, and Tz on the x, y, and z axis direction and a central rotation vector Rx, Ry, and Rz on the x, y, and z axis, respectively. Optimization of the transformation function is performed in the order of translation TT and rigid transformation TR using parameter sets T∈ℝ3, R∈ℝ6, respectively, as shown in the following equation [15]:(4)TTT =MTT13TRR =MTR13·MRR46MTT =100tx010ty001tz0001, MRθ =czcy−szcx+czsyszszsx+czsycz0szcyczcx+szsysz−czsx+szsycz0−sycysxcycx00001
where c=cosθ and s=sinθ. By using DICOM information, the displacement of the translation and rotation vector was limited in the transformation function as shown in Equation (5), so the execution time of registration was shortened. In addition, Powell’s optimization method [29] was used to converge to the optimal position. As such, it was possible to enable fast and accurate registration.
(5)Tx ≤ dmax, Ty ≤ dmax, Tz ≤ dmaxRx ≤ thR, Ry ≤ thR, Rz ≤ thR
where dmax and thR represent the threshold of translation and rotation vectors, respectively. These values were experimentally determined.

### 2.3. Graph Search and Analysis

In the case of 2D XA images, confusion regarding the background exists owing to the presence of organs, bones, and surgical instruments. In addition, as the images of actual 3D blood vessels are captured after being projected on a 2D plane, the depth information is unavailable. Consequently, there are errors in the structural information of the 2D centerlines that are segmented from the 2D XA image; moreover, these errors may cause errors in nonrigid registration. In this paper, to alleviate the occurrence of errors in nonrigid registration, the vascular structure in the 2D XA image was reconstructed based on the phase structure in the 3D CTA image and used in the nonrigid registration. Initially, a graph structure for the vascular structure was generated to perform an efficient 2D/3D vascular comparison by lowering the complexity of the vascular structure. This was represented as (*T, f*), consisting of a graph structure *T* and shape properties *f*. The graph structure *T* was composed of vertices υ, edge *E* (*E* ⊂ *V* × *V*), and root *r*, and was represented by *T* = (*V, E, r*). The shape properties *f* were composed of n centerline points per edge, and represented by *f*_3D_: *E* → *R*^3n^ and *f*_2D_: *E* → *R*^2n^.

In the case of the graph structure from the 3D vascular structure *T*_3D_, as bifurcation points were clearly identified and segmented, a tree structure was created. Conversely, in the case of the 2D vascular structure, false bifurcation points may occur owing to the confusion of the background. In addition, areas where blood vessels overlap or cross each other may occur because of the lack of depth information. To solve this problem, the graph structure for the 2D vascular structure was created for the smallest unit as shown in Figure 3.

For the reconstruction of the shape properties of the 2D vascular structure, i.e., *f*_2D_, this paper assumed the following two facts. (1) Because the 2D and 3D blood vessel data of the same patient were used, accurately segmented *f*_2D_ and *f*_3D_ were similar. (2) Through rigid registration, the same *f*_2D_ and *f*_3D_ existed within a certain range.

Based on these assumptions, we consider the set of candidate segments of *f^(i)^*_2D_ that satisfies the condition *d*_2_(*f^(i)^*_2D_*, f^(i)^*_3D_) < *D**^Th^* for each *f^(i)^*_3D_ that is defined as *S^(k)^_3D→2D_* (*k* = 1, 2, …, *r*), where *r* refers to the number of 2D candidate segments. To obtain the connectivity information that is similar to the vascular structure and shape of *f^(i)^*_2D_ and *f^(i)^*_3D_, the connected structure was generated using *S^(k)^_3D→2D_*. The connected structure created for *S^(k)^_3D→2D_* was defined as *L^(i)^_2D_*. In *L^(i)^_2D_*, all paths were searched to create all connectable structures among the candidate segments, as shown in Figure 4.

Finally, the connection between candidate lines, hereafter referred to as candi-lines, were analyzed to remove unnecessary candi-lines. The analysis of connection relations was performed for *T*_3D_ beginning from the lowest level including the leaf node toward the highest level including the root node. As the 3D vascular structure was a seamless connected structure, the connection elements of 2D candi-lines were analyzed based on the 3D segments connected to adjacent parent-child nodes. In this case, the child candi-lines that were not connected to the parent candi-lines were regarded as unnecessary candi-lines and removed.

### 2.4. Point Matching Using Distance and Gradient

The information on the curvature of the centerline was used to determine the optimal point in the 2D centerline that matches each point of the 3D centerline. In cases where point matching was performed using only simple curvatures [15], wrong point matching could be achieved in certain areas, as shown in Figure 5b, owing to variations such as errors, beats, and breathing that occurred during the segmentation or registration process. In addition, multiple points may be matched owing to the lack of information pertaining to the depths in 2D XA images. To perform robust and accurate matching in consideration of these errors, Equation (6) was calculated using the information on all the neighboring adjacent points, which were considered together.
(6)$$M_{2D}^{\left(i\right)}=argmin\left(\frac{1}{d_{2}}\frac{\left|c_{i}^{’}x_{j}-y_{j}+b\right|}{\sqrt{{\left(\frac{1}{2n+1}\sum_{i=-n}^{n}c_{i}^{’}\right)}^{2}+1}}+d_{2}\right)$$

Here, $c_{i}^{’}$ represents the vertical gradient of the *i*th point of the 3D centerline and (*x_j_, y_j_*) represents the 2D Point matched with (*x**_i_, y**_i_*). *n* denotes the number of adjacent points of (*x**_i_, y**_i_*) and *d*_2_ denotes the Euclidean distance between two points. *b* represents the intercept of the straight line that has a gradient of $c_{i}^{’}$ and passes through (*x_j_, y_j_*).

Figure 5c shows the results of point matching performed using the proposed method. As it can be observed in Figure 5c, the point matching method that uses the information of adjacent points is more uniformly and accurately performed than point matching using only the curvature or only one point.

### 2.5. Similarity Comparison

In this paper, the optimal candi-lines are selected through Equation (7) using the distances and differences in gradients, lengths, and thicknesses between 3D segments and candi-lines for accurate similarity comparison.
(7)$$SM=argmin\left(\alpha D^{\theta }+\beta d_{2}+\gamma D^{Th}+\delta D^{L}\right)$$

In Equation (7), *D^θ^*, *D^Th^*, and *D^L^* indicate the differences in gradients, thicknesses, and lengths between 3D segments and candi-lines, respectively. *α*, *β*, *γ*, and *δ* have values that are not smaller than zero, and indicate the degrees of influence on each term. They are determined experimentally. Each term is defined as follows.
(8)$$D^{\theta }=\sum \frac{c_{i}^{’}x-y+b}{d_{2}\sqrt{c_{i}^{’}{}^{2}+1}}$$
(9)$$D^{Th}=\frac{1}{N}\sum {\left(Th_{3D}-Th_{2D}\right)}^{2}$$
(10)$$D^{L}=\;\left|\left(S_{3D}^{L}-C_{2D}^{L}\right)/S_{3D}^{L}\right|$$

*Th*_3D_ and *Th*_2D_ represent the thicknesses of the blood vessels in the 3D and 2D segments, respectively, and s*^L^*_3D_ and c*^L^*_3D_ represent the lengths of the projected 3D segments and 2D candi-lines, respectively. The thicknesses of the 3D blood vessels are measured through the lengths of the vectors perpendicular to the gradient of the blood vessels. The lengths are constructed under the condition of $\left(v_{i},v_{i+1}\right)\;\in E$ at $i\in \;\left[1,\;k-1\right]$ for path *ρ* between the start and end points where the two vessels are matched.
(11)$$\rho \left(v,v^{\prime }\right)\;=\;\left(v=v_{1},v_{2},\cdots ,v_{k}=v^{\prime }\right)$$

The length of *ρ* is defined as the sum of the Euclidean distances of the matched points.
(12)$$d\left(\rho \right)\;=\overset{k-1}{\underset{i=1}{\sum }}\sqrt{{\left(v_{i}-v_{i+1}\right)}^{2}}$$

### 2.6. Nonrigid Deformation through Energy Minimization

Energy functions are defined to change the positions and shapes of the vascular centerlines; the positions of the 3D vascular centerline points are changed using the gradient descent method so that the energy functions have minimum values. The energy functions are defined as a combination of three terms, as presented in Equation (13).
(13)$$E=\overset{N}{\underset{i=0}{\sum }}\left(\mu E_{cont}\left(v_{i}\right)\;+\;\tau E_{curv}\left(v_{i}\right)\;+\;\varphi E_{image}\left(v_{i}\right)\right)$$

Each energy term performs the role of inducing the shape and position to approximate the 2D centerline while preserving the topological information of the 3D centerline. *E_cont_* denotes the continuity of the centerline. When a centerline vs. is composed of *N* pieces of points, *v*_1_, *v*_2_, … *v_N_*, the *E_cont_* at the *i*th point is calculated according to Equation (14).
(14)$$E_{cont}\left(v_{i}\right)\;={\left(\bar{d}\left\|v_{i}-v_{i-1}\right\|\right)}^{2}$$

Here, $\bar{d}$ denotes the average distance between adjacent points, which ensures that the distance between the adjacent points of the centerline is maintained constantly. *E_curv_* performs the role of ensuring that the centerline does not vibrate or suddenly changes in flexion; consequently, it creates a gentle and natural shape. *E_curv_* refers to the energy term for the calculation of curvature and is expressed as presented in the following equation.
(15)$$E_{curv}\left(v_{i}\right)\;={\left|\frac{d^{2}v_{i}}{ds^{2}}\right|}^{2}$$

*E_cont_* and *E_curv_* are the energy terms that are used to determine the basic structure of the centerline, while *E_image_* reflects the characteristics of the centerline. Thus, it is an energy term that changes the position and shape of the 3D blood vessel according to the 2D blood vessel.
(16)$$E_{image}\left(v_{i}\right)\;=H\left(v_{i}\right){\left(\left\|v_{i}-\dot{v_{i}}\right\|\right)}^{2}$$

*H*(*υ*) represents {0, 1}. When *H*(*υ*) = 0, it indicates that (1) there is no matching point or (2) the distance between the two points is more than the value of *Max_dist_*; for the remaining cases, *H*(*υ*) = 1 [15]. ${\dot{v}}_{i}$ represents the point matched to *v_i_*.

## 3. Experimental Results

The experiment in this study was implemented in C ++ and Python and was performed on a PC with Intel Core i7-8700 CPU (3.2 GHz), NVIDIA Geforce 1080 TI(11GB) 32 GB RAM and Windows 7, 64-bit operating system installed with Visual Studio 2017 and Python 3.5.

### 3.1. Dataset

In this study, data from a total of 68 patients were used, and the CTA images of both the left coronary artery (LCA) and right coronary artery (RCA) were used in the case of 27 patients, while the CTA images of either the LCA or RCA were used in the case of 11 patients. For the XA image, 394 series of images at different angles were captured from the 68 patients. Figure 6 shows a series of XA images with different angles for the same patient’s data.

For the CTA data, images were captured using Siemens SOMATOM Definition Flash, GE Lightspeed VCT, and Toshiba Aquilion ONE; moreover, the images had a resolution of 512 × 512 pixels and were captured at the end-diastole or end-systole. To convert the pixel units into physical units, values ranging from 0.305 to 0.39 and 0.5 to 0.75 were used, for pixel spacing and slice thickness, respectively. The XA images were captured using Philips Allura Xper, GE DL, and Toshiba DFP-8000D C-arm. The XA images had a resolution of 512 × 512 pixels acquired at a rate of 15 frames per second. These images included blood vessels that were visualized by administrating a contrast agent among consecutive frame images. For the pixel spacing, values in a range of approximately 0.258 to 0.293 were used.

For the experiments, 344 series of 394 series of XA images were used as training data, and the remaining 50 series were used as test data for evaluation. Ground truth for training of CNN segmentation was created by a cardiologist and researcher with more than three years of experience using the semi-automated blood vessel segmentation method [22] and Adobe Photoshop CS3. A total of 2771 training data were used through the data augmentation.

Ground truth data were created and used as a reference standard for quantitative evaluation of the accuracy of segmentation and registration. One frame was selected out of the 50 series of data as the ground truth, and a cardiologist and researcher with more than three years of experience manually segmented the coronary arteries from the original XA images using Adobe Photoshop CS3. The manually segmented ground truth was used as a basis for comparison for obtaining segmentation results in previous studies; therefore, a similar method was considered in the proposed method.

### 3.2. Simulation Data

In this study, for evaluating the accuracy of segmentation and registration, simulation data were generated prior to the evaluation through clinical data for performing a preliminary evaluation. Through the preliminary evaluation, the performance of the proposed method was characterized in ideal environments and scenarios and 3D registration errors, which cannot be evaluated with clinical data, were evaluated [15]. The simulation data were generated using a method similar to the method proposed by [15], as shown in Figure 7. Initially, a blood vessel segmented from a 3D CTA image was projected on an X-ray image in which no blood vessel was contrasted. In this case, to generate a natural image similar to an actual XA image, the Hounsfield unit information of the CT image was reflected when the blood vessel was projected. In addition, Gaussian smoothing and alpha blending were performed on the projected image for improvement and natural integration of the area around the boundary between two different images.

However, this method has a limitation that nonrigid deformation cannot be evaluated in cases where the same 3D CTA image is used for the creation and registration of simulation data. To improve this problem, 3D CTA images of the same patient that are captured on different dates or phases are used for the creation of simulation data and nonrigid registration. Using simulation data, parameters *α*, *β*, *γ*, and *δ* were used as 0.4, 0.7, 0.25 and 0.15 based on the optimal candi-line inclusion and the number of candidates. The parameters μ, τ, and φ were measured as error values as 0.5, 0.4 and 0.6 based on the average values of 3D Euclidean distance. For each parameter, one of the parameters was varied, the other two were kept fixed (as shown in Figure 8).

The simulation data had the characteristics that was contained in the conversion information, which was generated during creation, and were identical to the 3D blood vessel used for registration. These characteristics enabled accurate error measurement through the same correspondence point; moreover, it was possible to obtain 2D and 3D error measurements. Figure 9 and Figure 10 show the result of the accuracy measurement of the proposed method for the simulation data.

For the accuracy measurement of segmentation, the average values of 2D Euclidean distances were measured as error values for the 2D centerline/segment for which CNN-based segmentation with the centerline/segment of the simulation data was performed. The average error values of centerline pertaining to the LCA and RCA for 15 data values were measured to be 0.0975 and 0.0884 mm, respectively, and the average error values of segment were determined to be 0.6206 and 0.6195 mm, respectively. Segments were an important component of blood vessels and structures and were useful for understanding structures and nonrigid registration. For the accuracy measurement of registration, the average values of the 3D and 2D Euclidean distances projected on the X-ray image were measured as error values for the 3D centerline for which nonrigid registration with the centerline of the simulation data was performed. The 2D average error values pertaining to the LCA and RCA for 15 data values were measured to be 0.1374 and 0.0892 mm, respectively, and the 3D average error values were determined to be 0.6439 and 0.8615 mm, respectively. Therefore, the results of experiment performed using simulation data showed that the proposed method can derive robust and accurate results of segmentation and registration.

### 3.3. Clinical Data

#### 3.3.1. Segmentation of 2D XA Images

Figure 11 shows the results of the segmentation achieved by using the proposed method. Figure 11a,b shows the results of the segmentation for the LCA and RCA, respectively. It can be observed from Figure 11 that the blood vessels in the 2D XA images, which were captured during the procedure, segmented well and could be visually confirmed.

In this study, we compared the accuracy of the blood vessel segmentation of our method with those of the previous methods [10,24,30,31,32,33,34,35,36,37,38,39]. Table 2 presents the quantitative results of the eleven methods [24,30,31,33,34,35,36,37,38,39] for the segmentation of the coronary arteries on XA images, reporting the average similarity scores of 50 selected XA images with respect to the ground truth. Precision, recall, and F1 score were based on true positive (TP), false positive (FP), true negative (TN), and false negative (FN). TP represents the number of vessel pixels classified as vessel, FP represents the number of background pixels classified as vessel, TN represents the number of background pixels classified as background, and FN represents the number of vessel pixels classified as background. Precision, Recall, and F1 score represent TP/(TP+FP), TP/(TP+FP+FN), and 2TP/(2TP+FP+FN), respectively.

CRFasRNN [33], U-Net [24], and DeepLabV3+ [34] were the results of applying the segmentation method using a convolutional neural network. Frangi [30] and Krissian [31] were representative 2D blood vessel segmentation studies. DPWrenGA [35], LBAdaptiveSOM [36], MixtureOfGaussianV2 [37], MultiLayer [38], and PixelBasedAdaptiveSegmenter [39] were the background subtraction (BGS) methods [40] used in computer vision. We selected the top 5 best BGS methods [35,36,37,38,39]. We used the BGS library [40,41], and the parameter settings of each BGS method. Some previous methods showed high precision, but they resulted in very poor recall as a trade-off for increased precision. The recall was generally lower in classical image processing methods [30,31,35,36,37,38,39] than in CNN-based methods [24,33,34], which resulted in a very poor F1 score.

Table 3 presents the quantitative results of the three methods [10,32] for the segmentation and its structure of the coronary arteries, reporting the average similarity scores of 50 selected XA images. For the quantitative measurement of the segmentation and its structure results, the points and bifurcations of the same vessel between the segmentation result and ground truth were compared. When the points and bifurcations of the centerline of the segmentation result were within 1 and 3 mm with respect to the ground truth, it was defined as TP, respectively.

Table 3 shows that the proposed method is generally more accurate than the previous methods [10,32]. In particular, it can be seen that the proposed method was significantly higher in the accuracy of blood vessel bifurcations. As the bifurcation points of the blood vessels were important anatomical feature information that are required for the correct insertion of the surgical instrument during the coronary intervention, accurate positional information of the branching points should be provided to obtain highly reliable guidelines.

#### 3.3.2. Nonrigid Registration

Figure 12 shows the results of the registration achieved by using the proposed method. Figure 12a,b shows the results of the registration for the LCA and RCA, respectively. The color of the centerline in the registration results indicate the depths; for higher depths, the color s greener, while for lower depths, the color is redder, as shown in Figure 13. It can be observed from Figure 12 that the blood vessels in the 3D CTA image, which were captured before the procedure, and 2D XA images, which were captured during the procedure, matched well and could be visually confirmed.

To evaluate the accuracy of registration, errors were measured by comparing the results of the registration, which was automatically matched through the proposed method, with the ground truth centerline of the 2D XA image, which was manually segmented by experts. While measuring the distance error for the entire blood vessel, it is possible that the average error may be measured as a small value even when wrong blood vessels are matched. Consequently, in this paper, the robustness of the registration was further measured using bifurcation points and markers. As the bifurcation points of the blood vessels are important anatomical feature information that are required for the correct insertion of the surgical instrument during the coronary intervention, accurate positional information of the bifurcation points should be provided to obtain highly reliable guidelines. The points that were manually entered by experts for the 3D CTA image and ground truth centerline were used as the markers. The robustness of the registration was measured using the average of the distance errors between the markers.
(17)$$ADD=\frac{1}{M}\overset{M}{\underset{i=1}{\sum }}E\left(p_{i}-q_{i}\right)$$

Here, *p* and *q* denote the pairs of markers at two blood vessels and *E*(*p* − *q*) denotes the Euclidean distance between the corresponding markers. M indicates the number of markers; moreover, 10 pairs were used per patient.

The results of measurement of accuracy and robustness of the registration are shown in Figure 14.

Table 4 lists the results of the proposed method and previous studies. The accuracy evaluation indexes used in experiments differed according to the study, and there were significant differences in image modality. Therefore, there were limitations in determining clear and accurate criteria for comparison. However, the proposed nonrigid registration method showed an accuracy of less than 1 mm and an execution time of about 0.1 s. This indicates fast and accurate results.

Figure 15 shows the results of a nonrigid registration using the proposed method and method proposed by Kim et al. [16]. The average of distance difference for the centerline of the proposed method and that of the method proposed by Kim et al. [16] were 0.8705 and 1.2237 mm, respectively, which are numerically similar. However, the registration method proposed by Kim et al. [16] may lead to a final matching result in which branch points are not matched as shown in Figure 15 because certain blood vessels converge on the local minimum at points where there are local errors.

## 4. Discussion

XA image is commonly used in percutaneous coronary interventions with the advantage of visualization of the inside of blood vessels in real-time. However, the XA image acquisition process has the following drawbacks: (a) loss of depth information (b) background clutter of bones, organs, and intervention tools (c) fast-disappearing contrast agents (d) invisible areas such as lesions, etc. These drawbacks make it difficult to grasp and analyze the exact structure of blood vessels using XA images.

We approached in phase to provide accurate information from 2D XA images. CNN segmentation was performed to extract blood vessels from the XA image that is input at 15 frames per second at various angles. It was a structure with maximum accuracy within the constraints of expression capacity and computational complexity through compound scaling, and the prediction accuracy was improved with much fewer parameters and FLOPS through optimization. In addition, a loss metric considering the shape of blood vessels was learned. As a result, the overall blood vessel was rapidly and robustly extracted from the XA images acquired from various angles. Then, we utilized 3D vascular information to improve the fundamental drawbacks of 2D XA images. A graph structure was used for the efficient utilization of blood vessel information. Based on the 3D graph structure, the 2D graph structure was analyzed and reconstructed. In this process, errors caused by the background clutter were improved. In particular, a more accurate structure was extracted by improving the vascular area overlapped or obscured by the projection. Nonrigid registration was performed similarity comparisons and energy functions that preserve topology information in consideration of cardiovascular characteristics to complement 2D blood vessel information. The proposed method evaluated the entire blood vessel, markers, and bifurcation points. The markers were manually entered by experts. It was confirmed that the vascular structure was reconstructed and the nonrigid registration was performed correctly. In particular, there are cases in which blood vessels visible in 3D CTA are not visible in 2D XA images. The evaluation using markers is important for determining whether to apply to patients such as the cases (like total occlusion). Bifurcation points are used as important anatomical features for the accurate insertion of instruments. Therefore, it was evaluated for clinical applicability. In addition, the evaluation using phantom was used for error measurement in 3D. The proposed method demonstrated fast, robust, and accurate results for segmentation and nonrigid registration in clinical data.

The proposed segmentation method is useful for segmentation such as the shape of long and thin conduits such as vessels, and it is possible to optimize the structure using compound scaling according to the input image. The registration method is flexible to apply to other vessels composed of hierarchical structures, which includes the process of analyzing the structure based on the characteristic information of the blood vessels. In addition, since the entire registration process is performed based on the centerline extracted from the 2D/3D images, a dimensional expansion for 3D–3D registration is flexible.

## 5. Conclusions

In this paper, CNN-based segmentation and the nonrigid deformation method to represent the movement of the coronary artery in real-time while compensating for changes in the position and shape of the coronary artery due to other external factors such as heartbeat and breathing were proposed. The CNN-based segmentation extracted the robust and rapid results for 2D XA images which were obtained from various angles in real time. We also used 3D CTA information to complement the limitations caused by the projection of 3D into 2D. The phase structures between blood vessels extracted from 2D XA and 3D CTA images were analyzed to utilize the characteristics of data from the same patient with different pieces of modality information. Graph structures were used for efficient comparative analysis of the phase structure while reducing the complexity of the blood vessels. Although the phase structure of 3D blood vessels was clearly divided based on the bifurcation point, the phase structure of 2D blood vessels contained certain inaccurate information. To overcome this limitation, all possible connection paths were assumed to reconstruct the entire structure by comparatively analyzing the connection relations; further, point matching between two vessels was performed to verify and apply the accurate corresponding relations. The point matching was performed using distances and gradients together, and more accurate results were obtained by using information of adjacent points and not merely one point. Although the graph structure is efficient for calculation and comparative analysis as it lowers the complexity of blood vessels, thereby simplifying the blood vessels, it lacks feature information such as the curvature, thickness, and shape of blood vessels. Therefore, similarity comparison was performed to utilize the feature information. In addition, energy functions were defined and utilized based on the blood vessel feature information to reflect changes in the position and shape according to other external factors such as heartbeat and breathing so that registration to the optimal position was possible.

According to the results of the experiment, the error for the all the blood vessels was 0.8705 mm, the error for the marker, which was manually entered by the experts, was 1.06 mm, and the error for the bifurcation point was 1.5706 mm. In addition, the overall execution time of the segmentation and nonrigid registration was 0.179 s. Therefore, it can be concluded that an accurate segmentation and nonrigid registration was performed within a short amount of time.

## Figures and Tables

**Figure 1 diagnostics-12-00778-f001:**
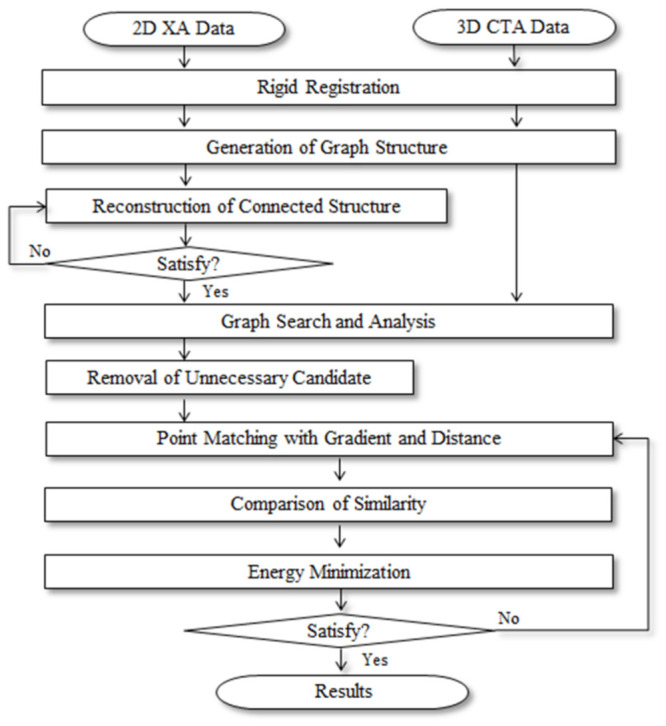
Process of proposed method.

**Figure 2 diagnostics-12-00778-f002:**
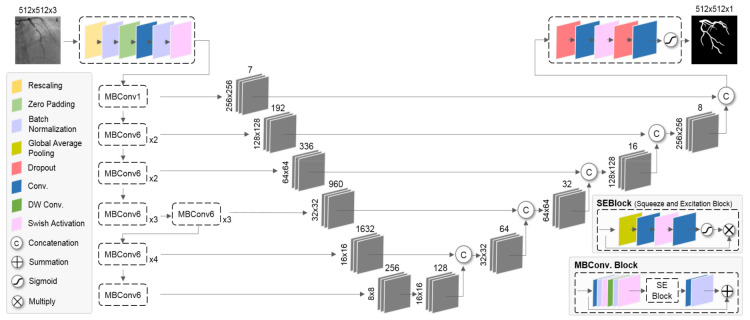
Overall network architecture of the proposed method.

**Figure 3 diagnostics-12-00778-f003:**
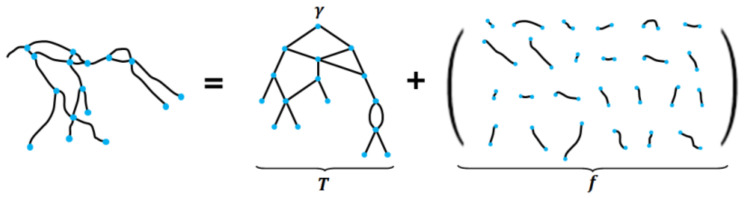
Creation of graph structure for vascular structure.

**Figure 4 diagnostics-12-00778-f004:**
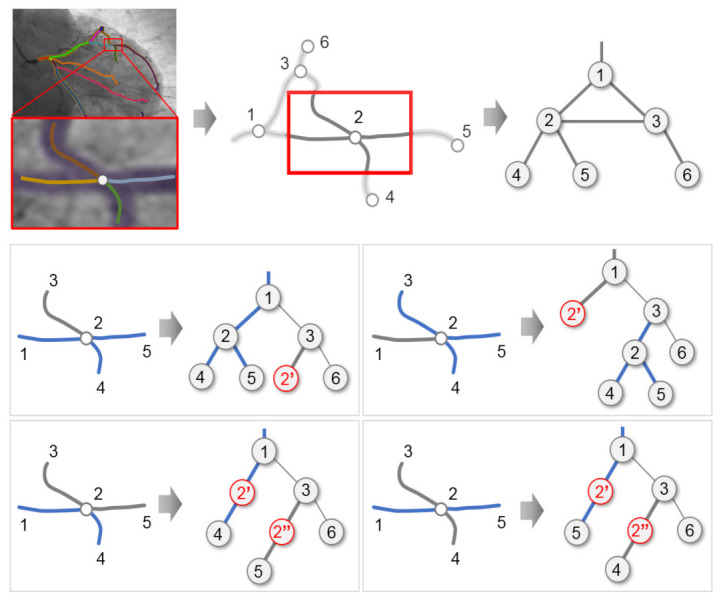
Creation of all connected structures that can be combined.

**Figure 5 diagnostics-12-00778-f005:**
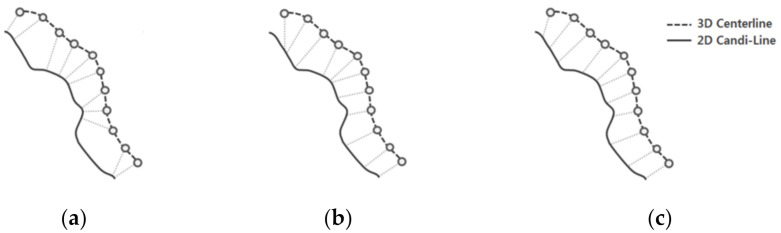
Comparison of point matching methods. (**a**) Distance-based methods. (**b**) Gradient-based methods. (**c**) Proposed method.

**Figure 6 diagnostics-12-00778-f006:**
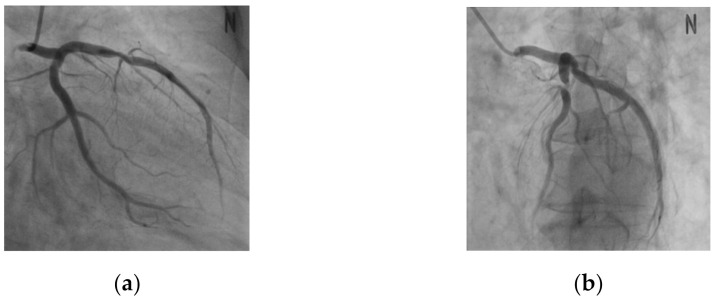
XA Images with different angles for the same patient. (**a**) Primary angle: −29.3, secondary angle: −18.7. (**b**) Primary angle: 42.6, secondary angle: 15.8.

**Figure 7 diagnostics-12-00778-f007:**
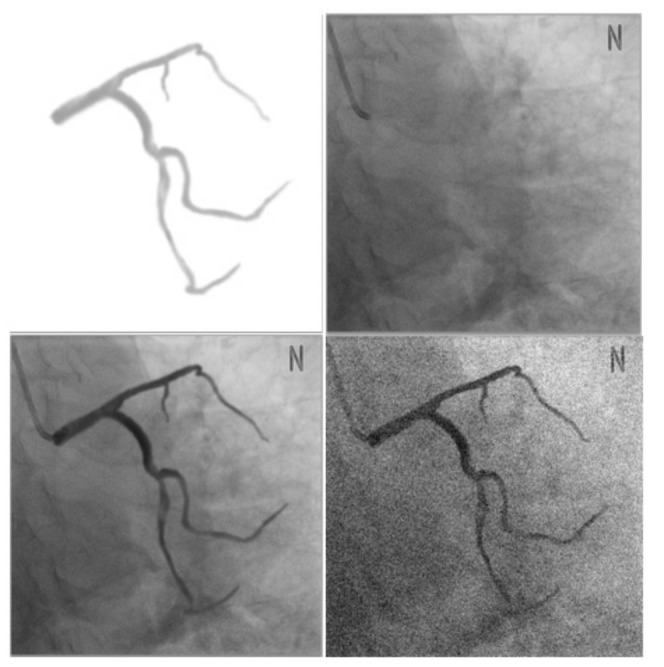
Creation of simulation data.

**Figure 8 diagnostics-12-00778-f008:**
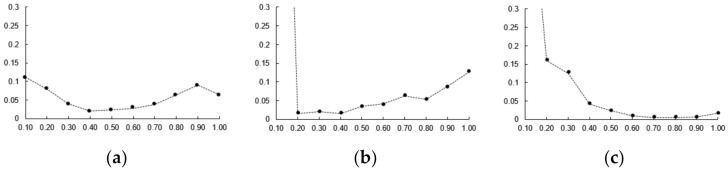
Error after nonrigid registration. (**a**) Parameter *μ*, (**b**) parameter *τ*, (**c**) parameter *φ*.

**Figure 9 diagnostics-12-00778-f009:**
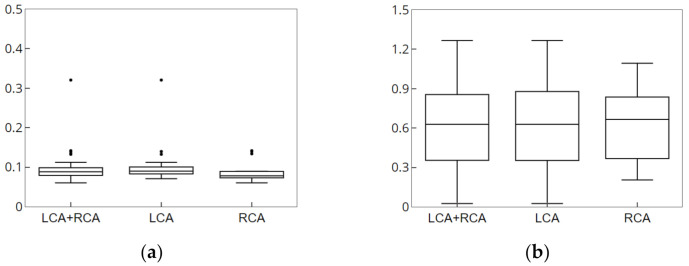
Result of segmentation of simulation data. (**a**) Error measurement of centerline. (**b**) Error measurement of segment.

**Figure 10 diagnostics-12-00778-f010:**
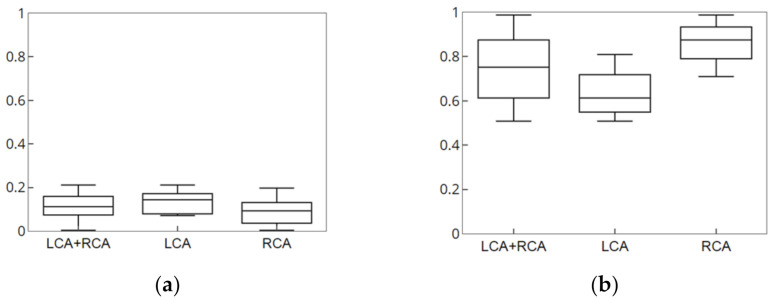
Result of nonrigid registration of simulation data. (**a**) Error measurement of 2D data. (**b**) Error measure of 3D data.

**Figure 11 diagnostics-12-00778-f011:**
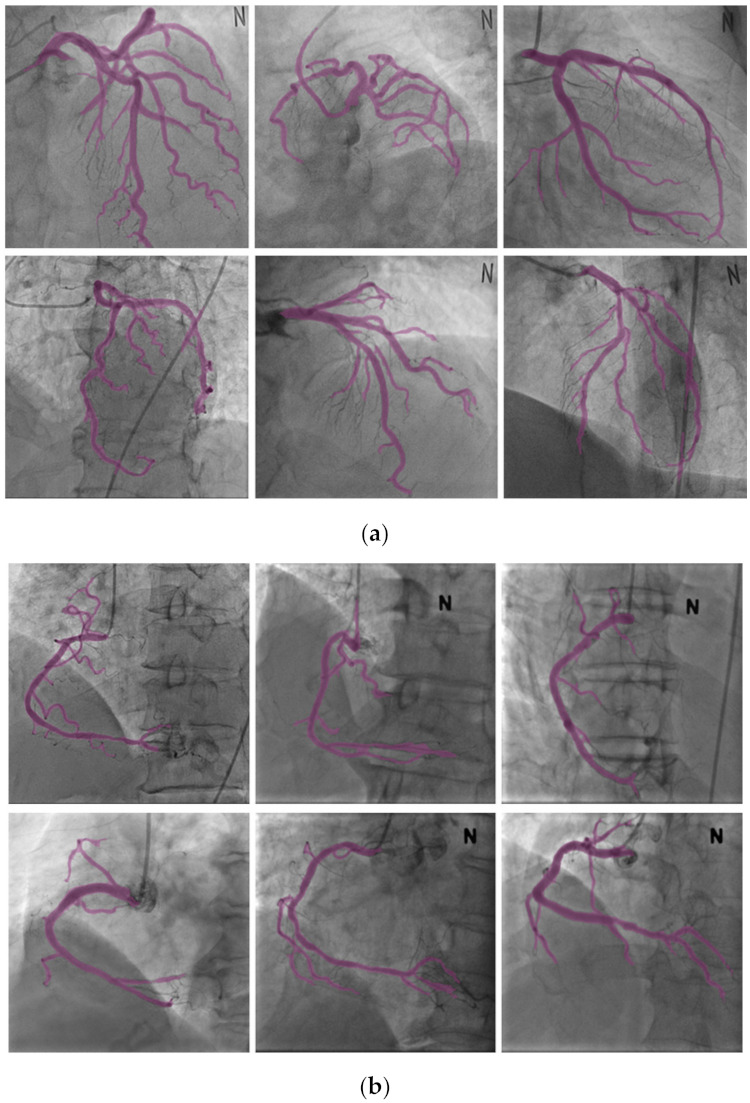
Results of segmentation. (**a**) Results of left coronary artery segmentation. (**b**) Results of right coronary artery segmentation.

**Figure 12 diagnostics-12-00778-f012:**
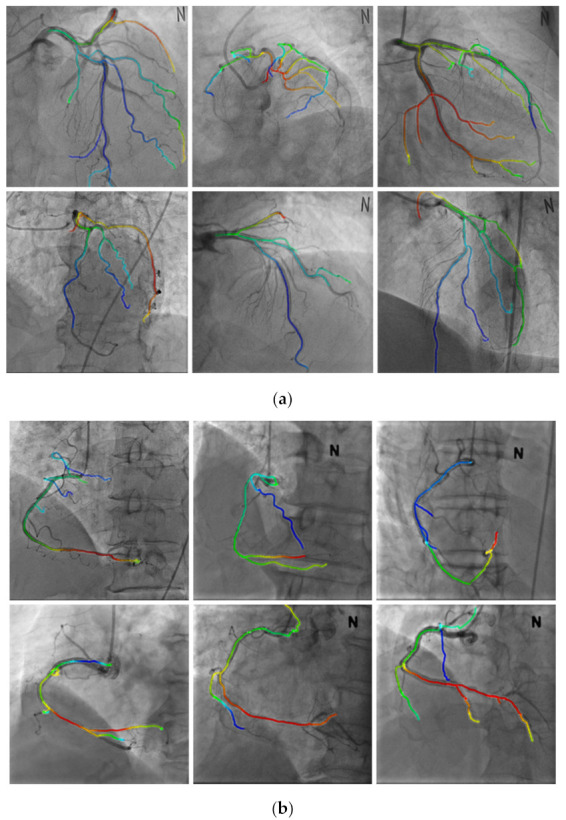
Results of nonrigid registration. (**a**) Results of left coronary artery registration. (**b**) Results of right coronary artery registration.

**Figure 13 diagnostics-12-00778-f013:**
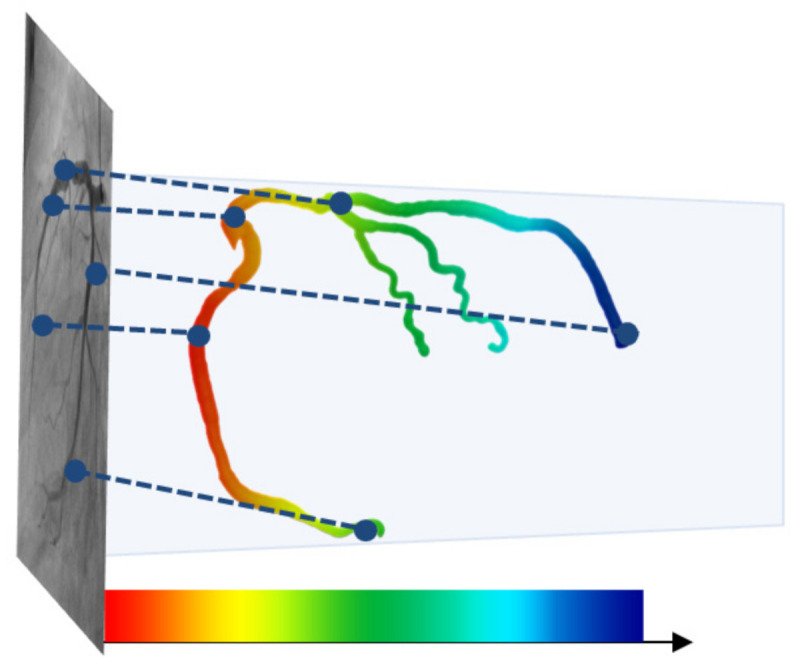
Color representation according to the depth of blood vessels [21].

**Figure 14 diagnostics-12-00778-f014:**
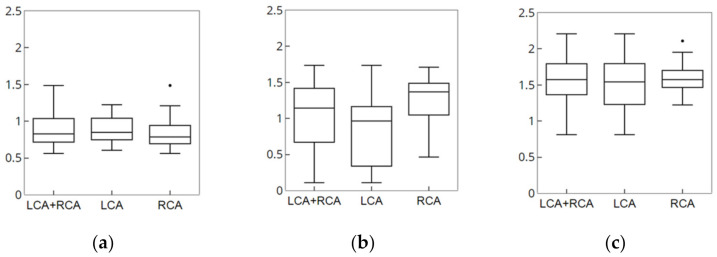
Results of measurement of the nonrigid registration of clinical data. (**a**) ADD. (**b**) Marker. (**c**) Bifurcation.

**Figure 15 diagnostics-12-00778-f015:**
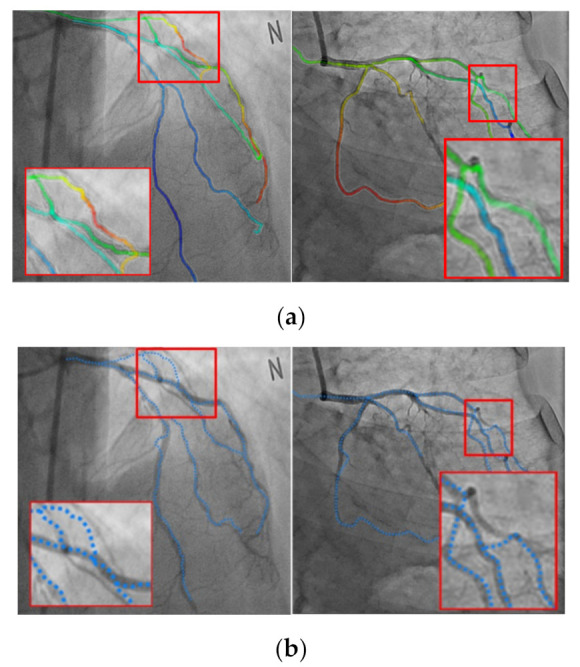
Comparison of the nonrigid registration results. (**a**) Proposed method. (**b**) Previous study [16].

**Table 1 diagnostics-12-00778-t001:** DICOM information [21].

Keyword	Tag
PositionerPrimaryAngle	(0018, 1511)
PositionerSecondaryAngle	(0018, 1510)
ImagerPixelSpacing	(0018, 1164)
DistanceSourceToDetector	(0018, 1110)
DistanceSourceToPatient	(0018, 1111)

**Table 2 diagnostics-12-00778-t002:** Results of comparison of the proposed method with previous studies [24,30,31,33,34,35,36,37,38,39].

Method	Precision	Recall	F1 Score
Proposed method	0.7563	0.6922	0.7176
CRFasRNN [33]	0.7378	0.5091	0.5874
U-Net [24]	0.7083	0.4713	0.5497
DeepLabV3+ [34]	0.6431	0.5266	0.5528
Frangi [30]	0.6725	0.3232	0.4172
Krissian [31]	0.2883	0.5078	0.3463
DPWrenGA [35]	0.3960	0.2582	0.2851
LBAdaptiveSOM [36]	0.4016	0.2526	0.2822
MixtureOfGaussianV2 [37]	0.5713	0.1325	0.2051
MultiLayer [38]	0.4631	0.1811	0.2128
PixelBasedAdaptiveSegmenter [39]	0.7784	0.1435	0.2250

**Table 3 diagnostics-12-00778-t003:** Results of comparison of the proposed method with previous studies [10,32].

Method	Points			Bifurcations		
	Precision	Recall	F1 Score	Precision	Recall	F1 Score
Proposed method	0.8542	0.6003	0.7035	0.8897	0.6389	0.7386
Sun et al. [10]	0.6009	0.2575	0.3235	0.5637	0.2018	0.2738
Park et al. [32]	0.6974	0.3737	0.4475	0.7235	0.4152	0.4708

**Table 4 diagnostics-12-00778-t004:** Results of the proposed method and previous studies.

Author	Modality	Dimensionality	No. of Test Subjects	Accuracy	Processing Time (s)
Kaila et al. [14]	Biplane XA/CTA	3D–3D	7	1.41 mm (RMSE ^1^)	N/A
Khoo et al. [42]	Biplane XA/CTA	2D–3D	6	3.8 mm (RMSD ^2^) 2.31 mm (RMSD ^2^)	0.4 15
Liu et al. [43]	XA/CTA	2D–3D	10	0.6201 pix (MPE ^3^)	20
Park et al. [44]	XA Sequence	2D–2D	9	7.02 mm (TRE ^4^)	N/A
Kim et al. [16]	XA/CTA	2D–3D	12	4.25/5 (Score ^5^)	4.5
Proposed method	XA/CTA	2D–3D	50	0.8705 mm (ADD ^6^)	0.107

^1^ RMSD: Root mean square distance. ^2^ RMSE: Root mean square error. ^3^ MPE: Mean projective error. ^4^ TRE: Target registration error. ^5^ Score: Average score by three experts on a five-point scale. ^6^ ADD: Average of distance difference.

## Data Availability

Not applicable.

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
