# Peer review of "Deep Learning Segmentation in 2D X-ray Images and Non-Rigid Registration in Multi-Modality Images of Coronary Arteries"

_diagnostics, 2022, doi:10.3390/diagnostics12040778_

Round 1

Reviewer 1 Report

Title : Deep Learning Segmentation in 2D X-ray Images and Nonrigid Registration in Multi-modality Images of Coronary Arteries
Authors : Taeyong Park, Seungwoo Khang, Heeryeol Jeong, Kyoyeong Koo, Jeongjin Lee, Juneseuk Shin and Ho Chul Kang
-----------------------
In this paper, the authors present a novel segmentation and nonrigid registration method to provide useful real-time assistive images and information, with an overall process execution time inferior to 200ms. 

Strength: 
. The methodology section (2) is very interesting and remarquably illustrated. 

Weakness:
. The experimental results section could be more developed in your revision. 

Originality / Novelty:
. This problem is original, the question is well defined and the idea proposed in this paper is interesting. 

Significance:
. The scientific content of this paper is correct for me and deserves to be published. 
. The hypotheses are correctly identified as such. 
. The presented  results are appropriately presented and significant, even if they could gain in being clarified in the revised version. 
. The technical quality of this paper is correct for me. 
. The conclusion is correctly justified and supported by relevantly discussed results. 
. Some limits of the results obtained in this paper are mentioned. This point could be more developed to improve the paper overall quality.

Quality of presentation:
. The abstract is clear and presents correctly the subject addressed in this paper. 
. This paper contains the basic sections of a scientific paper. More, the use of subheadings makes it clearer and easier to understand. It is clear, easy to follow and to read, and logically written. 
. The data and experimental analyses could be better presented. 
. The conclusion is argumented and clear enough. 
. The English language quality and style of this paper are appropriate and understandable.

Scientific soundness:
. The subject addressed in this paper is relevant.
. The study has been correctly designed, and is technically sound. 
. The analysis of the results could gain of a more detailed investigation. 
. I took interest and pleasure to read this paper.
. In my opinion, methods and conclusions of this paper seem to be interesting for the readership of the journal. 

References :
. 46 research references, out of which 3 self-references, giving an acceptable self-reference ratio equal to 6.5%. The chosen references are overall relevant.
. Avoid citing groups of references: [1-4], [5-7], [9-12], [13-19], [32-41], [35-41], [37-41]. Or if you do it anyway, please comment more any of these references. 
. The references are all cited in the text, but would deserve to be more deeply analysed with regard to the addressed subject. 
. The bibliography of this paper is mainly composed of recent references: 18 of them are more than 10 years old, and 28 of them are less than 10 years old.
. Please avoid the formulation 'et al.' in the references section: [5], [7]. The complete list of authors deserve to be cited in this section. 

Overall evaluation:
. I think there is an overall benefit to publish this work. 
. This work provides an advance towards the current knowledge, clearly highlighted in the abstract. 
. The authors have addressed an already studied question, but with smart experiments as well as a correct bibliography. 
As a conclusion, my suggestion to the editor is to accept this paper for publication in diagnostics.

Author Response

Response to the Reviewers’ comments;

Reviewer # 1 (Comments and Suggestions for Authors)

In this paper, the authors present a novel segmentation and nonrigid registration method to provide useful real-time assistive images and information, with an overall process execution time inferior to 200ms.

1. Avoid citing groups of references: [1-4], [5-7], [9-12], [13-19], [32-41], [35-41], [37-41]. Or if you do it anyway, please comment more any of these references.

Response) Thanks for your constructive suggestion. Revised as suggested to the manuscript. 

2. The references are all cited in the text, but would deserve to be more deeply analysed with regard to the addressed subject.

Response) Revised as suggested to the manuscript with Comment #1.

3. Please avoid the formulation 'et al.' in the references section: [5], [7]. The complete list of authors deserve to be cited in this section.

Response) Thanks for your detailed comment. Revised as suggested to the manuscript.

As a conclusion, my suggestion to the editor is to accept this paper for publication in diagnostics.

Response) Thank you for your decision, sincerely.

Reviewer 2 Report

This is an interesting manuscript, in which the authors propose a new method for semantic segmentation and non-rigid deformation of the coronary artery in real time. The experiments appear solid and convincing. My only suggestion is to re-read the manuscript as I have encountered some typos and repetitions. Finally, the data should be made public to allow for reproducibility by others.

Author Response

Response to the Reviewers’ comments;

Reviewer # 2 (Comments and Suggestions for Authors)

This is an interesting manuscript, in which the authors propose a new method for semantic segmentation and non-rigid deformation of the coronary artery in real time. The experiments appear solid and convincing.

1. My only suggestion is to re-read the manuscript as I have encountered some typos and repetitions.

Response) Thanks for your constructive suggestion. Revised as suggested to the manuscript.

2. Finally, the data should be made public to allow for reproducibility by others.

Response) Thanks for your constructive suggestion. Data used in this study was provided to the authors with Institutional Review Board (IRB) approval from Severance Hospital. Data cannot be sent to outside in accordance with hospital policy as it has been accepted for the purpose of this study only. Therefore, the data are not publicly available.